# Increased Risk of Malignancy with Immunosuppression: A Population-Based Analysis of Texas Medicare Beneficiaries

**DOI:** 10.3390/cancers15123144

**Published:** 2023-06-11

**Authors:** Luca Cicalese, Jordan R. Westra, Casey M. O’Connor, Yong-Fang Kuo

**Affiliations:** 1Division of Transplant Surgery, Department of Surgery, University of Texas Medical Branch, Galveston, TX 77555, USA; 2Office of Biostatistics, University of Texas Medical Branch, Galveston, TX 77555, USA; 3Department of Biostatistics & Data Science, University of Texas Medical Branch, Galveston, TX 77555, USA; 4Sealy Center on Aging, University of Texas Medical Branch, Galveston, TX 77555, USA

**Keywords:** immunosuppressive drugs, immunosuppression, cancer, liver cancer, skin cancer, lymphoma, kidney cancer, transplantation

## Abstract

**Simple Summary:**

This study analyzed patients receiving immunosuppressive drugs (IMD) for the prevention of organ transplant rejection or treatment of other conditions (rheumatoid arthritis, lupus, myasthenia gravis, interstitial lung disease, fibromyalgia, and other autoimmune diseases). This retrospective study utilized Medicare claims data from all Texas Medicare beneficiaries between 2007 and 2018. In these patients, the risk of developing cancer was evaluated. We found an increased risk of cancer for all patients using IMD, regardless of its indication or duration, with a distribution of the types of cancer different from that previously described and with a higher risk for liver cancer. We also observed a higher risk of cancer in younger patients and ethnic minorities.

**Abstract:**

Immunosuppressive drugs (IMD) are widely utilized to treat many autoimmune conditions and to prevent rejection in organ transplantation. Cancer has been associated with prolonged use of IMD in transplant patients. However, no detailed, systematic analysis of the risk of cancer has been performed in patients receiving IMD for any condition and duration. We analyzed Medicare data from Texas Medicare beneficiaries, regardless of their age, between 2007 and 2018, from the Texas Cancer Registry. We analyzed the data for the risk of cancer after IMD use associated with demographic characteristics, clinical conditions, and subsequent cancer type. Of 29,196 patients who used IMD for a variety of indications, 5684 developed cancer. The risk of cancer (standardized incidence ratio) was particularly high for liver (9.10), skin (7.95), lymphoma (4.89), and kidney (4.39). Patients receiving IMD had a four fold greater likelihood of developing cancer than the general population. This risk was higher within the first 3 years of IMD utilization and in patients younger than 65 years and minorities. This study shows that patients receiving IMD for any indications have a significantly increased risk of cancer, even with short-term use. Caution is needed for IMD use; in addition, an aggressive neoplastic diagnostic screening is warranted.

## 1. Background

In recent years, improvements in existing immunosuppressive drugs (IMDs) and the development of new and more potent ones have extended their use to a variety of medical conditions [1,2,3,4,5]. Despite the wider utilization of these medications, their effect on the development of cancer has not been carefully analyzed. Historically, most of the patients receiving IMD have been organ transplant recipients. Such patients require lifelong use of these medications to prevent rejection, enhance graft survival, and improve quality of life [6,7,8]. Despite the substantial benefits in terms of quality of life and mortality, the result is exposure to prolonged reduced immune surveillance. This, in addition to certain infections that increased with IMD use, has been associated with a higher risk of malignancies [9,10,11]. Many cancers themselves appear to be associated with viral infections observed more often when patients are immunosuppressed, such as lymphomas (Epstein–Barr virus), cervical cancer (human papillomavirus), Kaposi’s sarcoma (human herpes virus), and liver cancer (human hepatitis virus B and C) [12,13]. Moreover, other relationships, such as acquired cystic disease and subsequent renal cell carcinoma, may be related to factors associated with the prolonged period of chronic uremia observed specifically in kidney transplant recipients while waiting for a suitable graft to become available [14].

However, other cancers not associated with viral infections or other specific conditions of renal insufficiency might also present in immunosuppressed patients. The risk of developing any type of cancer has not been deeply evaluated, nor is it known in this patient population. While the prolonged use of IMD in transplant patients has been associated with an increased risk of malignancy, the risk of cancer associated with short-term exposure to IMD has not been previously evaluated.

Moreover, many IMDs are available to treat conditions other than transplantation and are now widely used for a variety of non-transplant related indications, such as RA, lupus, myasthenia gravis, interstitial lung disease, fibromyalgia, and several other autoimmune diseases. The risk associated with these treatments is not well known.

Additionally, cancer in Texas might have a peculiar incidence/distribution of certain cancers that could vary when compared to other states in the US. For example, the incidence of liver cancer (hepatocellular carcinoma) varies widely among states, with Texas having one of the highest incidence rates. Texas is reported to rank third in the US, with an incidence rate approximately double the national rate [15]. Texas is home to the second largest petrochemical industry and agricultural industry in the nation and therefore its population is exposed, in certain areas, to higher levels of potentially dangerous pollutants. When we specifically analyzed the hypothesis that chronic exposure to certain pollutants can increase the risk of such cancer, we found a higher incidence of liver cancer in Texas counties with higher air levels of various chemical pollutants [16]. We have also observed a consistent, significant, positive association between the incidence of liver cancer and hepatitis C prevalence rates with vinyl chloride concentrations [17]. Moreover, we previously described that proximity to an oil refinery in Texas is associated with an increased risk of various types of cancer [18].

All these factors might make Texas different when compared to other states in the US or internationally, and warrant a specific analysis of the effects of IMD on cancer peculiar to this population, information that is currently not available.

## 2. Methods

### 2.1. Data Source

This retrospective study utilized Medicare claims data from all Texas Medicare beneficiaries between 2007 and 2018. Medicare files used for this study included the Master Beneficiary Summary File (MBSF), Medicare Provider Analysis and Review (MedPAR) files, Outpatient Standard Analytic (OutSAF) files, Medicare Carrier files, and Part D Event (PDE) files. Information on beneficiary demographics and enrollment was taken from the MSBF; diagnoses and procedures were taken from MedPAR (inpatient services), OutSAF (institutional outpatient services), and Carrier (services from providers); and prescription drug information was taken from the PDE files. 

Cancer cases for the general population of Texas were obtained from the Texas Cancer Registry (TCR) for the years 2008 (to include one year of IMD usage) to 2018. TCR data contains demographic information on cancer patients as well as cancer-specific information, such as the tumor site, histology, and behavior. Estimates for the population of Texas were obtained from the American Community Survey (ACS) 1-year data for the years 2010 to 2018. Texas population estimates for 2008 and 2009 were obtained from the ACS 5-year estimates from 2010, due to the unavailability of the 1-year data for those years. The University of Texas Medical Branch Institutional Review Board waived the requirement for informed consent for this study using de-identified patient data.

### 2.2. Cohort

The Medicare cohort consisted of those who initiated the use of IMD between 2008 and 2017, had continuous Medicare coverage (parts A, B, and D with no health maintenance organization enrollment) in the 12 months before initiation, and had no cancer diagnosis code in the 12 months before initiation. IMD use included the use of two types of drugs: those which are mainly used among organ transplant patients and those mainly used for other conditions (e.g., RA, psoriasis, lupus, inflammatory bowel disease). Transplant IMD included tacrolimus, sirolimus (rapamycin), cyclosporine, and mycophenolate. IMD for other conditions included abatacept, adalimumab, alefacept, anakinra, azathioprine, basiliximab, belatacept, canakinumab, certolizumab, daclizumab, etanercept, golimumab, infliximab, leflunomide, muromonab-cd3, omalizumab, rilonacept, teriflunomide, tocilizumab, tofacitinib, ustekinumab, and vedolizumab. IMD use was identified from the Healthcare Common Procedure Coding system (HCPCS) from the OutSAF or Carrier files, and from the National Drug Code (NDC) from Part D. The comparison group consisted of all Texas cancer patients as the numerator (from the TCR) and all Texas residents as the denominator (from the ACS). 

### 2.3. Measures

The primary outcome was a new diagnosis of cancer. For the Medicare cohort, this was determined by International Classification of Disease, Ninth or Tenth Revision (ICD-9 or ICD-10) codes found in any MedPAR, OutSAF, or Carrier claims files (ICD-9: 140.x-172.x, 173.0, 173.10, 173.19, 173.20, 173.29, 173.30, 173.39, 173.40, 173.49, 173.50, 173.59, 173.60, 173.69, 173.70, 173.79, 173.80, 173.89, 173.9, 173.90, 173.99, 174.x-208.x, 209.0, 209.1, 209.2, 209.30, 209.31, 209.32, 209.33, 209.34, 209.35, 209.36, 209.7, 225.x, 227.3, 227.4, 228.1, 228.2, 230.x-234.x, 237.0, 237.1, 237.5, 237.6, 237.9, 238.4, 238.7, 239.6, 239.7, 273.3, 277.89; ICD-10: C00.x-C43.x, C44.0, C44.10, C44.19, C44.20, C44.29, C44.30, C44.39, C44.40, C44.49, C44.50, C44.59, C44.60, C44.69, C44.70, C44.79, C44.80, C44.89, C44.9, C44.90, C44.99, C45.x-C96.x, D00.x-D09.x, D18.2, D32.x, D33.x, D35.2, D35.3, D35.4, D42.x, D43.x, D44.3, D44.4, D44.5, D45, D46.x, D47.1, D47.2, D47.3, D47.4, D47.9, D49.6, D49.7, R85.614, R87.614, R87.624). For the TCR data, inclusion in the data was sufficient to indicate cancer diagnosis. Additionally, the type of cancer was identified and categorized as bladder, breast, colorectal, kidney, liver, lung, lymphoma, ovarian/uterine, pancreas, prostate, sarcoma, skin, or thyroid. All other cancer types were grouped together as “other.” In the TCR data, cancer type is determined by a combination of site, histology, and behavior codes. 

The variables of interest used to assess the association with cancer diagnosis included year of IMD initiation (2008–2013 vs. 2014–2018), sex, age at IMD initiation (0–34, 35–64, 65+ years), race (Black, Hispanic, White, Other), time since IMD initiation (<3 years, 3–5 years, 5+ years), and the presence of either organ transplantation (yes/no), end-stage renal disease (ESRD) diagnosis (yes/no), rheumatoid arthritis (RA) diagnosis (yes/no), or diabetes diagnosis (yes/no). Organ transplant was defined by procedure codes from MedPAR, OutSAF, and Carrier files (ICD-9-V3: 00.91, 00.92, 00.93, 07.94, 11.6, 11.60, 11.69, 33.50, 33.5, 33.51, 33.52, 33.6, 37.51, 41.00, 41.0, 41.01, 41.02, 41.03, 41.04, 41.05, 41.06, 41.07, 41.08, 41.09, 41.91, 41.94, 46.97, 49.74, 50.5, 50.51, 50.59, 52.80, 52.8, 52.82, 52.83, 52.84, 52.85, 52.86, 55.6, 55.69, 63.53, 65.92, 82.56, 82.58, 83.75, 83.77, 86.64; ICD-10-PCS: first position 0 and third position Y; HCPCS: 32850, 32851, 32852, 32853, 32854, 32855, 32856, 33927, 33928, 33929, 33930, 33933, 33935, 33940, 33944, 33945, 38240, 38241, 38242, 44135, 44136, 47133, 47135, 47136, 47140, 47141, 47142, 47143, 47144, 47145, 47146, 47147, 48160, 48550, 48551, 48552, 48554, 48556, 50300, 50320, 50323, 50325, 50327, 50328, 50329, 50340, 50360, 50365, 50370, 50380). Other comorbid conditions were taken from the Elixhauser comorbidity index in the 12 months before IMD initiation from MedPAR, OutSAF, or Carrier files and were measured as yes/no.

### 2.4. Statistical Analysis

Descriptive characteristics were calculated for the overall IMD cohort and for IMD users who developed cancer, reporting the N (%) for categorical variables and mean (standard deviation (SD)) for continuous variables. To assess the association of risk factors with the development of cancer among IMD users, a Fine and Gray proportional hazards model for competing events was used, with time to cancer diagnosis as the dependent variable and death as a competing event. Beneficiaries were censored if they lost Medicare coverage or died, or at the end of the study period (31 December 2018). Variables of interest included in the model: year of IMD initiation, sex, age category, race, organ transplant, ESRD, RA, and diabetes. The proportionality of hazards was assessed by entering the interaction term of the organ transplant and the log of the event time in the model. Hazards ratios (HR) and 95% confidence intervals (CI) were generated for each risk factor.

For comparison with the general population, the IMD users were compared to the TCR data using standardized incidence ratios (SIR). SIRs were calculated by comparing the observed number of cancer cases with the expected number, which was generated by multiplying the TCR cancer incidence rate in each year-, sex-, age-, and race-specific stratum by the person-years at risk. SIRs were calculated for each stratum of year of IMD initiation, sex, age category, race, and transplant status by time since IMD initiation, ESRD/transplant status (neither; ESRD only; transplant only; both), liver disease, RA, and diabetes. In addition to these strata, SIRs were calculated for specific cancers, including bladder, breast, colorectal, kidney, liver, lung, lymphoma, ovarian/uterine, pancreas, prostate, sarcoma, skin, and thyroid. Further reported were 95% CIs for each SIR. All analyses were performed using SAS (version 9.4, Cary, NC, USA).

## 3. Results

In our study cohort of 29,196 Medicare enrollees who initiated IMD use between 2008 and 2017 (Table 1), the mean age was 64.6 (SD = 15.2) years, 72.3% were women, 74.8% were white, 56.2% had an original disability or ESRD entitlement, and 44.5% were eligible for Medicaid. The major indications for IMD use included RA (49.8%), organ transplantation (15.8% total; 12.2% had both organ transplant and ESRD), psoriasis (8.1%), lupus (7.7%), and inflammatory bowel disease (5.1%). About 60% of patients had more than four comorbid conditions. Hypertension (78.9%), diabetes (40.2%), chronic obstructive pulmonary disease (35.8%), and hypothyroidism (30.6%) were the four most prevalent conditions. When followed until the end of 2018, 5684 had a cancer diagnosis. The most common cancers in our study population were skin (10.6%), lung (9.3%), breast (8.6%), lymphoma (6.4%), and colorectal (6.0%) (Table 2).

In the multivariable analyses, organ transplantation was associated with an increased risk of cancer. The adjusted cancer incidence rate at 10 years was 47.8% for those with a transplant and 26.5% for those without a transplant. The risk of cancer diagnosis was not constant over time, thus violating the proportional hazards assumption (*p* < 0.0001). Therefore, based on the divergence of the hazard plot, we estimated the risk of cancer by organ transplant status for three periods of time: <3 years, 3 to <5 years, and 5+ years. The risk of cancer associated with transplant decreased with time, with a hazard ratio of 2.16 (95% CI: 1.92, 2.44) in the first 3 years after IMD initiation, 2.00 (95% CI: 1.65, 2.43) for between 3 and 5 years, and 1.84 (95% CI: 1.45, 2.33) at 5 or more years after IMD initiation. The demographic factor associated with a higher risk of cancer diagnosis was older age (HR: 1.46, 95% CI: 1.24, 1.73 for those aged 35–64 years and HR: 2.46, 95% CI: 2.08, 2.91 for those aged 65+ years, compared to those aged < 35 years). Demographic factors associated with a lower risk of cancer diagnosis included being female (HR: 0.75, 95% CI: 0.71, 0.80), and being a minority (HR: 0.80, 95% CI: 0.72, 0.88 for Black and HR: 0.76, 95% CI: 0.66, 0.89 for Hispanic, compared to white). RA and diabetes were not statistically significantly associated with an increased risk of cancer diagnosis (Table 3). 

Compared to the general population, patients with IMD use had a more than four-fold greater likelihood of a subsequent cancer diagnosis (SIR: 4.39, 95% CI: 4.27, 4.50). In our study, almost all types of cancer were statistically significantly higher in the IMD population than in the general population, except for sarcoma. The risk of cancer was particularly high for liver (SIR: 9.10, 95% CI: 7.86, 10.34), skin (SIR: 7.95, 95% CI: 7.32, 8.59), lymphoma (SIR: 4.89, 95% CI: 4.39, 5.39), and kidney (SIR: 4.39, 95% CI: 3.81, 4.96) cancers, all organ sites previously identified as being at risk of cancer in transplant patients on chronic immunosuppression. However, the risk of other types of cancer (i.e., ovarian/uterine, thyroid, colorectal, pancreas, prostate, and bladder) was also more than 2.5 times greater (Table 4) for patients taking IMD compared to the general population. 

The association between IMD and cancer was stronger among patients younger than 35 years (SIR: 73.19, 95% CI: 51.59, 84.79), among those aged 35–64 years (SIR: 6.90, 95% CI: 6.54, 7.26), among men (SIR: 4.96, 95% CI: 4.74, 5.18), and among minority populations (Black: SIR: 4.55, 95% CI: 4.15, 4.96; Hispanic: SIR: 5.33, 95% CI: 4.55, 6.11). The risk of cancer associated with IMD use was also higher for transplant patients, regardless of ESRD status (with ESRD: SIR: 9.93, 95% CI: 9.22, 10.65; without ESRD: SIR: 9.62, 95% CI: 8.61, 10.62). Transplant patients also saw a higher risk of cancer in the first three years of IMD initiation (SIR: 10.36, 95% CI: 9.68, 11.03) compared to non-transplant IMD users (SIR: 4.09, 95% CI: 3.95, 4.22). Compared to their counterparts without these conditions, the association between IMD use and cancer was higher for patients with liver disease (SIR: 5.42, 95% CI: 5.06, 5.77) and those with diabetes (SIR: 4.87, 95% CI 4.67, 5.07). This relationship was not found for patients with RA (Table 4).

## 4. Discussion

The overall risk of developing cancer in patients receiving chronic IMD treatment after organ transplantation has been previously identified to be 2.6 times that of the general population [19]. This increase was associated with reduced immunosurveillance against tumors and with an increased number of viral infections, seen with a higher frequency in these patients and known to be associated with certain specific cancers. However, there is limited knowledge of the effect of IMD on non-transplant indications or the effect of a less extensive duration of use on the development of cancer.

Therefore, this study analyzed for the first time this entire patient population for the State of Texas, identifying all patients receiving IDM for any indication, concluding that not just transplant patients, but all patients using IMD, have a significantly increased risk of developing many types of cancers. 

In patients using IMD, as expected, the rates of overall cancer diagnosis increased with age, peaking in those 65 years and older compared to those of younger age. However, compared to the general population, the association between IMD and cancer was stronger among patients younger than 35 years, becoming progressively less in those aged 35–64 years, and less still in those 65 years and older. 

Most patients receiving IMD were white (74.8%), while the rest were Black (10.7%), Hispanic (11.0%) and other (3.6%). This ethnic distribution is roughly that described in the 2020 Census for the population in Texas (white 61.6%, Black 12.4%, Hispanic 18.7%, other 7.3%), allowing for the lower use of healthcare in non-white populations [20]. However, analysis based on race of the actual rate of cancer versus the expected rate indicated that ethnic minorities are exposed to a much higher risk of developing cancer than white patients when using IMD.

These higher rates of cancer in younger and Black patients could be explained in transplant patients in part by the higher amount of IMD usually needed in these patients after organ transplant, due to their more robust immune activity and higher risk of rejection, with the potential risk of excessively lowering the immune defense [21]. However, this explanation is only speculative, since the more robust immune system of these groups would also suggest better immunosurveillance (e.g., against tumor cells). Moreover, these groups are associated overall with reduced compliance (younger patients) and reduced absorption of certain IMD drugs (calcineurin inhibitors in Black patients), leaving them with a more pronounced immune surveillance activity, confirmed by the increased rate of graft rejection seen in these patients after transplant [22,23]. Additionally, this phenomenon is even more difficult to explain in patients who received no transplant but did receive a shorter course of IMD treatment, and in Hispanic and Asian patients.

We identified clinically important associations between IMD use, cancer, and co-morbidities. Patients receiving IMD are often medically complex, and our study cohort showed a high rate of co-morbidities (about 60% of patients had more than four co-morbidities). Compared to those without these conditions, the association between IMD use and cancer was higher for patients with ESRD, liver disease, and diabetes. Similar associations have been previously described in the general population [24,25,26,27]. However, this association was not found for patients with ESRD when the population was controlled for transplant and in those with RA (Table 3). Liver disease is often associated with viral infections such as hepatitis B and C or with alcohol abuse, and the association of these conditions with hepatocellular carcinoma is well established [12]. In our analysis, the risk of cancer was particularly high for the liver (SIR 9.10, 95% CI: 7.86, 10.34). Among the different states, Texas has been shown to have a relatively high rate of liver cancer in the general population. The national incidence rate of liver cancer (hepatocellular carcinoma) per 100,000 persons age-adjusted to the 2000 US population was reported to be 5.4, with a mortality rate of 2.1. Comparatively, Texas ranks third in the US, with an incidence almost double the national rate (9.9 per 100,000) and a mortality rate over 3 times the national rate [15]. We hypothesize that such a high rate observed in Texas could be linked to the combination of known individual risk factors (hepatitis and alcohol abuse) with specific exposure to environmental factors. Therefore, we analyzed a large number of known chemical pollutants regularly measured and recorded in Texas and found an association between the presence of certain clusters of pollutants with liver cancer [16], as well as a consistently significant positive association between the incidence of liver cancer and hepatitis C prevalence rates with the concentrations of vinyl chloride [17]. Although this specific subset of immunosuppressed patients was not analyzed by geographical distribution or by association with the level of pollutant distribution, the use of IMDs appears to potentiate the risk further for liver cancer. Furthermore, the suppressive effect of these drugs on the immune system could affect specific immunomodulatory defensive pathways naturally preventing liver cancer. 

The overall type and frequency of cancers diagnosed in our patient analysis are summarized in Table 2. Previous studies have identified the association of certain cancers with transplant patients with chronic IMD use [28,29,30,31,32]. The risk of these factors has been reported as high for skin cancer (>10-fold increase), Kaposi sarcoma (>50-fold increase), non-Hodgkin’s lymphoma (>8-fold increase), and cancers of the anogenital tract (vaginal, cervical, vulval, anal, and penile cancer; >4-fold increase). Other cancers have been reported at rates only slightly higher than those expected in the general population: colorectal, ~40% increase; melanoma, double risk; and lung cancer, >50% increase. Other cancers also increased, including head and neck, thyroid, esophagus, stomach, leukemias, and plasma cell tumors. Cancers such as breast cancer and prostate cancer did not show an increase in risk for organ transplant recipients [33]. However, our study identified a different pattern of cancer risk from those previously reported, including one particularly high for the liver, as discussed, but also for skin, lymphoma, kidney, and prostate. Naturally, as discussed above, this result may be peculiar to Texas, because of the greater exposure to environmental pollutants affecting the liver or other organs or the more prolonged sun exposure during the year for skin cancer, or for other unidentified reasons. However, those taking IMD faced a 2.5 times greater risk of ovarian/uterine, thyroid, colorectal, pancreas, and bladder cancer compared to patients exposed to the same factors but not taking IMD. This risk was found in patients taking IMD for indications other than organ transplants. Clearly, additional analysis is needed to tease out these associations.

Notably, our data included both transplant patients on chronic immunosuppression as well as patients on a shorter IMD treatment course for other indications. In fact, our data suggests that the risk is highest within the first 3 years of initiating IMD treatment and progressively drops with time. Organ transplantation was associated with a 2-fold higher risk of cancer compared to IMD use for other indications. This result can be explained by the more potent immunosuppression used in transplantation compared to IMD used for other indications. Specific IMD drugs used in transplantation include tacrolimus, sirolimus (rapamycin), cyclosporine, potent T-lymphocyte inhibitors acting on interleukin-2, and mycophenolate which is a potent T and B-lymphocyte cytostatic agent. Basically, these inhibit the activity and cloning of T and B cells in response to donor antigen presentation. We believe that the difference in the effect seen in this study between transplant patients (i.e., receiving these specific IMD) and other non-transplant patients can be explained by a difference in potency or mechanism of actions of different IMD agents. In fact, IMD used for non-transplant use are mostly antibodies designed to target pre-formed lymphocytes. Hence, this difference could have a different effect on cancer immunosurveillance. A granular analysis of the effect of individual drugs should be performed to evaluate this effect in detail. Moreover, in transplantation, it is common to use a potent immunosuppressive induction with polyclonal or monoclonal antibodies (directed to reduce T and B cell populations) early after transplant, followed by high doses of IMD early on. Moreover, graft rejection is experienced more frequently early after transplant, requiring at times specific antirejection treatments with strong IMD. Only after 1 to 2 years post-transplant are these doses progressively reduced to a low-maintenance immunosuppressive dose. Additional studies may be able to determine the optimal immunosuppressive dose that also has the lowest risk of future cancer incidence. 

This study has some limitations. First, prescriptions reflect what was prescribed, not what the patient used, and cannot fully guarantee treatment adherence. Second, prescription data do not include diagnosis codes; therefore, the indications of prescription might not be completely captured from medical claims files. Third, socioeconomic factors, environment, and health behaviors could be important confounders of the association of race/ethnicity with cancer incidence in our SIR analyses. Fourth, the severity of comorbid conditions, which could modify the risk of cancer, is not available from Medicare data. Fifth, we did not examine the influence on the risk of cancer of the dosage, duration, or type of immunosuppression medication used. Finally, our findings may not be generalized to patients who were not fee-for-service Medicare Part D enrollees or to patients living outside of Texas.

However, this study analyzes in detail and for the first time the effect of IMD use for any indication (organ transplant as well as other autoimmune conditions) on the development of cancer.

## 5. Conclusions

In conclusion, our study in this population indicates an increased risk of cancer for all patients using IMD, regardless of its indication or duration. We also observed a higher risk in younger patients and white people, with a distribution of the types of cancer different from that previously described, and with a higher risk for liver cancer. Therefore, this study suggests the need for a granular analysis of IMD use and the risk of developing cancer at a national level. Clinically, caution is called for in the use of IMD treatments for indications different than transplantation, even when used for a short duration. A more careful selection of agents (especially for non-transplant patients), screening for preexisting liver conditions that can increase the risk of liver cancer, and the development of an aggressive and widened neoplastic diagnostic screening are strongly encouraged. 

## Figures and Tables

**Table 1 cancers-15-03144-t001:** Characteristics of immunosuppression drug (IMD) users.

	Cancer		All IMD Users	
	N	%	N	%
All	5684	100%	29,196	100%
IMD year				
2008	760	13.4%	2994	10.3%
2009	654	11.5%	2728	9.3%
2010	617	10.9%	2525	8.6%
2011	603	10.6%	2673	9.2%
2012	573	10.1%	2569	8.8%
2013	535	9.4%	2785	9.5%
2014	634	11.2%	3331	11.4%
2015	510	9.0%	3123	10.7%
2016	473	8.3%	3275	11.2%
2017	325	5.7%	3193	10.9%
Sex				
Male	1978	34.8%	8099	27.7%
Female	3706	65.2%	21,097	72.3%
Age, years				
<35	153	2.7%	1544	5.3%
35–44	220	3.9%	1926	6.6%
45–54	500	8.8%	3201	11.0%
55–64	693	12.2%	4135	14.2%
65–74	2434	42.8%	11,047	37.8%
75–84	1418	24.9%	5854	20.1%
85+	266	4.7%	1489	5.1%
Beneficiary Race				
White	4566	80.3%	21,831	74.8%
Black	487	8.6%	3117	10.7%
Other	180	3.2%	1043	3.6%
Hispanic	451	7.9%	3205	11.0%
Original Reason for Entitlement				
Age	2198	38.7%	12,783	43.8%
Disability/ESRD	3486	61.3%	16,413	56.2%
Dual Eligibility				
No	3780	66.5%	16,204	55.5%
Yes	1904	33.5%	12,992	44.5%
Elixhauser Count				
0	316	5.6%	1566	5.4%
1	460	8.1%	2600	8.9%
2	617	10.9%	3447	11.8%
3	681	12.0%	3675	12.6%
4	645	11.3%	3366	11.5%
5	598	10.5%	3049	10.4%
6	548	9.6%	2743	9.4%
7	439	7.7%	2175	7.4%
8	379	6.7%	1708	5.9%
9	297	5.2%	1341	4.6%
10+	704	12.4%	3526	12.1%
Transplant	1095	19.3%	4618	15.8%
ESRD	837	14.7%	4187	14.3%
ESRD/Transplant				
Neither	4493	79.0%	23,949	82.0%
Transplant only	354	6.2%	1060	3.6%
ESRD only	96	1.7%	629	2.2%
Both	741	13.0%	3558	12.2%
Rheumatoid arthritis	2766	48.7%	14,538	49.8%
Psoriasis	429	7.5%	2377	8.1%
Lupus	394	6.9%	2241	7.7%
Inflammatory bowel disease	309	5.4%	1485	5.1%
Alcohol abuse	98	1.7%	473	1.6%
Blood loss anemia	207	3.6%	982	3.4%
Cardiac arrhythmia	1707	30.0%	7514	25.7%
CHF	1180	20.8%	5601	19.2%
Coagulopathy	594	10.5%	2403	8.2%
COPD	2038	35.9%	10,444	35.8%
Deficiency anemia	1393	24.5%	6839	23.4%
Depression	1258	22.1%	7401	25.3%
Diabetes	2319	40.8%	11,743	40.2%
Drug abuse	102	1.8%	643	2.2%
Fluid and electrolyte disorders	1701	29.9%	8429	28.9%
HIV/AIDS	27	0.5%	90	0.3%
Hypertension	4615	81.2%	23,028	78.9%
Hypothyroidism	1834	32.3%	8925	30.6%
Liver disease	877	15.4%	4327	14.8%
Obesity	947	16.7%	5432	18.6%
Other neurological disorders	687	12.1%	3862	13.2%
Paralysis	102	1.8%	805	2.8%
Peptic ulcer disease, excluding bleeding	140	2.5%	784	2.7%
Peripheral vascular disorders	1352	23.8%	6420	22.0%
Psychoses	207	3.6%	1190	4.1%
Pulmonary circulation disorders	392	6.9%	1852	6.3%
Renal failure	1693	29.8%	8142	27.9%
Valvular disease	1160	20.4%	5089	17.4%
Weight loss	581	10.2%	2847	9.8%

ESRD: end-stage renal disease; CHF: congestive heart failure; COPD: chronic obstructive pulmonary disease; HIV: human immunodeficiency virus; AIDS: acquired immune deficiency syndrome.

**Table 2 cancers-15-03144-t002:** Distribution of cancer types.

Type	N	%
Breast	487	8.6%
Kidney	223	3.9%
Liver	208	3.7%
Lung	527	9.3%
Lymphoma	363	6.4%
Prostate	333	5.9%
Sarcoma	24	0.4%
Skin	603	10.6%
Other	2916	51.3%
Total	5684	100.0%

**Table 3 cancers-15-03144-t003:** Risk of cancer diagnosis after IMD initiation by characteristics.

Characteristics		Hazard Ratio (95% CI)
Year	2008–2013	REF
	2014–2018	1.03 (0.95, 1.06)
Age, years	0–34	REF
	35–64	1.46 (1.24, 1.73)
	65+	2.46 (2.08, 2.91)
Sex	Male	REF
	Female	0.75 (0.71, 0.80)
Race	White	REF
	Black	0.80 (0.72, 0.88)
	Hispanic	0.76 (0.66, 0.89)
	Other	0.69 (0.62, 0.76)
Transplant	<3 Years	2.16 (1.92, 2.44)
Yes vs. No	3 to <5 Years	2.00 (1.65, 2.43)
By time since IMD initiation	5+ Years	1.84 (1.45, 2.33)
ESRD	No	REF
	Yes	0.82 (0.72, 0.92)
Rheumatoid arthritis	No	REF
	Yes	1.06 (1.00, 1.12)
Diabetes	No	REF
	Yes	1.04 (0.98, 1.10)

IMD: immunosuppression drug; CI: confidence interval; ESRD: end-stage renal disease.

**Table 4 cancers-15-03144-t004:** Standardized incidence ratios.

		Cancer Cases	N	Person-Years	Expected Cases	SIR (95% CI)
Overall		5684	29,196	94,574.3	1295.2	4.39 (4.27, 4.50)
Year	2008–2013	2178	16,274	35,350.5	467.3	4.66 (4.47, 4.86)
	2014–2018	3506	22,526	59,223.8	828.0	4.23 (4.09, 4.37)
Cancer Type	Bladder	145	29,196	108,794.5	52.2	2.78 (2.33, 3.23)
	Breast	487	29,196	107,757.2	304.9	1.60 (1.46, 1.74)
	Colorectal	343	29,196	108,282.2	120.6	2.84 (2.54, 3.14)
	Kidney	223	29,196	108,577.5	50.8	4.39 (3.81, 4.96)
	Liver	208	29,196	108,781.8	22.9	9.10 (7.86, 10.34)
	Lung	527	29,196	108,365.2	224.7	2.35 (2.15, 2.55)
	Lymphoma	363	29,196	108,315.9	74.3	4.89 (4.39, 5.39)
	Ovarian/Uterine	124	21,097	77,019.4	26.3	4.72 (3.89, 5.55)
	Pancreas	128	29,196	108,976.3	45.8	2.80 (2.31, 3.28)
	Prostate	304	8099	30,903.7	110.0	2.76 (2.45, 3.07)
	Sarcoma	24	29,196	109,096.0	15.2	1.58 (0.95, 2.21)
	Skin	603	29,196	107,448.5	75.8	7.95 (7.32, 8.59)
	Thyroid	83	29,196	108,918.7	24.5	3.39 (2.66, 4.12)
Age	<35	153	1544	5049.8	2.1	73.19 (61.59, 84.79)
	35–64	1413	9262	31,193.9	204.7	6.90 (6.54, 7.26)
	65+	4118	18,390	58,330.6	1088.4	3.78 (3.67, 3.90)
Sex	Male	1978	8099	26,580.9	399.0	4.96 (4.74, 5.18)
	Female	3706	21,097	67,993.4	896.2	4.14 (4.00, 4.27)
Race	Black	487	3117	10,388.3	107.0	4.55 (4.15, 4.96)
	Hispanic	180	1043	3526.4	33.8	5.33 (4.55, 6.11)
	Other	451	3205	10,889.3	34.7	13.00 (11.80, 14.20)
	White	4566	21,831	69,770.3	1119.8	4.08 (3.96, 4.20)
Transplant	No	4589	24,578	81,374.3	1040.9	4.41 (4.28, 4.54)
	Yes	1095	4618	13,200.0	85.8	12.77 (12.01, 13.52)
Time Since IMD Initiation—Transplant	<3 Years	900	4618	9944.4	86.9	10.36 (9.68, 11.03)
	3 to <5 Years	115	1229	1880.9	15.0	7.68 (6.28, 9.09)
	5+ Years	80	602	1374.7	9.6	8.36 (6.52, 10.19)
Time Since IMD Initiation—No Transplant	<3 Years	3539	24,578	58,559.9	865.9	4.09 (3.95, 4.22)
	3 to <5 Years	638	8511	13,495.1	190.6	3.35 (3.09, 3.61)
	5+ Years	412	4155	9319.3	127.2	3.24 (2.93, 3.55)
ESRD	No	4847	25,009	82,266.6	1065.9	4.55 (4.42, 4.68)
	Yes	837	4187	12,307.7	66.8	12.52 (11.67, 13.37)
ESRD/Transplant	Neither	4493	23,949	79,600.3	1167.9	3.85 (3.73, 3.96)
	Transplant only	354	1060	2666.4	36.8	9.62 (8.61, 10.62)
	ESRD only	96	629	1774.1	15.9	6.02 (4.82, 7.23)
	Both	741	3558	10,533.6	74.6	9.93 (9.22, 10.65)
Liver Disease	No	4807	24,869	81,346.14	1133.3	4.24 (4.12, 4.36)
	Yes	877	4327	13,228.19	161.9	5.42 (5.06, 5.77)
Rheumatoid Arthritis	No	2918	14,658	46,214.1	607.3	4.80 (4.63, 4.98)
	Yes	2766	14,538	48,360.3	687.9	4.02 (3.87, 4.17)
Diabetes	No	3365	17,453	58,982.2	818.9	4.11 (3.97, 4.25)
	Yes	2319	11,743	35,592.1	476.4	4.87 (4.67, 5.07)

SIR: standardized incidence ratio; CI: confidence interval; IMD: immunosuppression drug; ESRD: end-stage renal disease.

## Data Availability

The data underlying this article were provided by the Centers for Medicare and Medicaid Services and the Texas Cancer Registry under Data Use Agreements. Data requests should be made to these entities.

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
