# Peer review of "Increased Risk of Malignancy with Immunosuppression: A Population-Based Analysis of Texas Medicare Beneficiaries"

_cancers, 2023, doi:10.3390/cancers15123144_

Round 1
Reviewer 1 Report
in the current manuscript, the authors analyzed Medicare data from Texas Medicare beneficiaries, and found patients receiving IMD had a four-fold greater likelihood of developing cancer than the general population. And that patients receiving IMD for any indications have a significantly increased risk of cancer, even with short-term use. While the analysis in current study is suitable and the information obstained is somewhat interesting to the general public and policy markers, i feel the paper is not a good fit for the topic "Inflammatory and Immunological Markers in Liver Cancers".
The English is well written in the manuscript.
Author Response
We have extended in the manuscript the liver-specific issues and discussion.
We hope the reviewer will consider our revisions with favor.
Reviewer 2 Report
This manuscript aims to statistically analysis of the causative relationship between immunosuppression and increased risk of malignancy. Specifically, the authors analyzed patients with immunosuppressive drug administration and their likelihood of getting some cancers. Although immunosuppression leading to cancer is expected, analyzing it using patient cases is quite interesting and can provide more statistical support. This manuscript can be recommended for publication after some minor revision. First, please indicate what the “SIR” and “CI” stand for (in line 19-20), when they first appear in the paper, especially in the abstract. Because the abstract should be independent, and readers want to know the meaning of abbreviations without reading the whole paper. Second, in the result section, the authors write out lots of data from patients, without giving short interpretations and conclusions of these data. Indeed, it is the interpretation and conclusions that give your data significance and attract readers to seek more interpretation in the discussion section. Third, this manuscript used lots of abbreviations and some of them are even not professional. For example, in the line 226, the “ETOH” should be “ethanol abuse” and ETOH is not a professional expression. It would be better if authors can define abbreviations when they first appear in the main sections, including abstract, introduction, methods, results, and discussion. In this case, readers do not need to find the phrases for the abbreviations and start reading any of those sections independently.
English is fine, but authors should reduce the use of abbreviation or define them clearly when they first appear in the manuscript.
Author Response
This manuscript aims to statistically analysis of the causative relationship between immunosuppression and increased risk of malignancy. Specifically, the Please find:
indicate what the “SIR” and “CI” stand for (in line 19-20), when they first appear in the paper, especially in the abstract. addressed, abbreviations spelled out
Second, in the result section, the authors write out lots of data from patients, without giving short interpretations and conclusions of these data. Indeed, it is the interpretation and conclusions that give your data significance and attract readers to seek more interpretation in the discussion section. we have added additional explanations in the results section
hird, this manuscript used lots of abbreviations and some of them are even not professional. For example, in the line 226, the “ETOH” should be “ethanol abuse” and ETOH is not a professional expression. Addressed with abbreviations modified and spelled out when first appear in the text.
Moreover we increased significantly the length of the article and revised the references in accordance with MDPI requirements
Round 2
Reviewer 1 Report
The authors analyzed the use of IMD on the future occurence of cancer. Their data come from the Medicare of the state of Texas which is considered a reliable source. The study found an increased risk of cancer for IMD usage even for patients without organ transplantation.
The following points must be modified to improve the manuscript:
1. The reasons why IMD usage caused increased risk of cancer is not discussed in depth.
2. Their exists multiple IMD drugs, the use of different IMD drugs should be analyzed.
3. The tables need to be redesigned to meet publication criteria.
Author Response
The reasons why IMD usage caused increased risk of cancer is not discussed in depth:
we added additional explanations in the text
Their exists multiple IMD drugs, the use of different IMD drugs should be analyzed.
we agree. transplant immunosuppression is different than non transplant immunosuppression and this is one of the reason of the different results observed in these groups. We elaborated in the text additional discussion on these results.
The tables need to be redesigned to meet publication criteria.
tables have been checked to make sure meet criteria and they are also submitted in a separate file to avoid formatting changes
